# Evolutionary histories and antimicrobial resistance in *Shigella flexneri* and *Shigella sonnei* in Southeast Asia

Hao Chung The [1], Ladaporn Bodhidatta[2], Duy Thanh Pham[1,3], Carl J. Mason [2], Tuyen Ha Thanh[1], Phat Voong Vinh[1], Paul Turner [3,4], Sopheak Hem[5], David A. B. Dance [3,6,7], Paul N. Newton[3,6,7], Rattanaphone Phetsouvanh[3,6,10], Viengmon Davong[6], Guy E. Thwaites[1,3], Nicholas R. Thomson [7,8], Stephen Baker [9] & Maia A. Rabaa [1,3✉]

Conventional disease surveillance for shigellosis in developing country settings relies on serotyping and low-resolution molecular typing, which fails to contextualise the evolutionary history of the genus. Here, we interrogated a collection of 1,804 *Shigella* whole genome sequences from organisms isolated in four continental Southeast Asian countries (Thailand, Vietnam, Laos, and Cambodia) over three decades to characterise the evolution of both *S. flexneri* and *S. sonnei*. We show that *S. sonnei* and each major *S. flexneri* serotype are comprised of genetically diverse populations, the majority of which were likely introduced into Southeast Asia in the 1970s–1990s. Intranational and regional dissemination allowed widespread propagation of both species across the region. Our data indicate that the epidemiology of *S. sonnei* and the major *S. flexneri* serotypes were characterised by frequent clonal replacement events, coinciding with changing susceptibility patterns against contemporaneous antimicrobials. We conclude that adaptation to antimicrobial pressure was pivotal to the recent evolutionary trajectory of *Shigella* in Southeast Asia.

[1] Oxford University Clinical Research Unit, Ho Chi Minh City, Vietnam. [2] Department of Enteric Diseases, Armed Forces Research Institute of Medical Sciences, Bangkok, Thailand. [3] Centre for Tropical Medicine and Global Health, Nuffield Department of Clinical Medicine, University of Oxford, Oxford, UK. [4] Cambodia-Oxford Medical Research Unit, Angkor Hospital for Children, Siem Reap, Cambodia. [5] Medical Biology Unit, Institut Pasteur du Cambodge, Institut Pasteur International Network, Phnom Penh, Cambodia. [6] Lao-Oxford-Mahosot Hospital-Wellcome Trust Research Unit, Microbiology Laboratory, Mahosot Hospital, Vientiane, Laos. [7] London School of Hygiene and Tropical Medicine, London, UK. [8] The Wellcome Trust Sanger Institute, Hinxton, Cambridge, UK. [9] The Department of Medicine, University of Cambridge, Cambridge, UK. [10]Deceased: Rattanaphone Phetsouvanh. ✉email: mrabaa@oucru.org

Shigella remains a common cause of dysentery in young children in developing countries, causing an estimated 160,000 deaths per year across all age groups globally[1,2]. Though composed of four species (*S. dysenteriae*, *S. boydii*, *S. flexneri*, and *S. sonnei*), >90% of shigellosis cases globally are attributed to the latter two species[3]. Regional epidemiological surveillance in Southeast Asia (SEA) has been important in characterizing the burden and distribution of different *Shigella* species and serotypes[4–7]. Such studies have highlighted the gradual decline of *S. flexneri* and emerging predominance of *S. sonnei*, particularly associated with multi-drug resistant (MDR) variants. However, conventional microbiological approaches cannot provide an in-depth understanding of the pathogens' population structures and recent life histories.

The use of whole genome sequencing (WGS) provides unprecedented granularity for studying the circulation, population dynamics, and evolution of a specific pathogen. An early key genomic study on the global collection of extant *S. sonnei* revealed that the contemporary variant of this pathogen likely emerged in Europe at least 400 years ago[8]. Among the extant lineages (1–5), Lineage 3 has been the most successful, with global dissemination in the 1980s and frequent association with antimicrobial resistance (AMR). WGS and genomic analyses also detailed how Lineage 3 *S. sonnei* evolved and propagated following its introduction into Southern Vietnam[9]. This study also suggested that resistance to contemporaneous antimicrobials was crucial in shaping the clonal expansion of *S. sonnei*. Similarly, a genomic investigation of ~350 *S. flexneri* genomes delineated seven phylogenetic groups constituting the species[10], excluding serotype 6 (which is phylogenetically closely related to *Shigella boydii* serotypes)[11]. Notably, these phylogenetic groups are inconsistent with serotype-based groupings and arose on multiple occasions between the 14th and 19th centuries. The distinct accumulation of virulence factors and antimicrobial resistance (AMR) determinants in *S. flexneri* Lineage 3 likely account for enhanced virulence and the international dominance of serotype 2a, the predominant serotype in Lineage 3[10].

Despite the discoveries made from these detailed molecular epidemiology studies, much remains unknown about the evolutionary history of *S. sonnei* and *S. flexneri* across SEA. Here, we aimed to better understand these relationships by performing WGS on a large historical collection of both *S. sonnei* and *S. flexneri* isolated across continental SEA. We aimed to investigate the population structures and evolution of these organisms, as well as to examine the establishment of AMR and the population dynamics of the two species in this region.

## Results

**Southeast Asian Shigella in global context**. We compiled a collection of 1804 *Shigella* genome sequences from across mainland SEA (Thailand, Vietnam, Laos and Cambodia), including 803 *S. sonnei* and 1001 *S. flexneri* isolates. Figure 1 details the geographical distribution of these samples in the region, with Bangkok, Central Thailand, and Southern Vietnam accounting for over half of the collection (*n* = 915). The correlation between the proportion of species isolated over time to those sequenced in this study in the well-sampled urban area of Thailand (Bangkok and Central provinces) over three decades (normalised cross-correlation >0.95 for both *S. flexneri* and *S. sonnei*; Supplementary Fig. 1) suggests this collection is likely to be representative of the epidemiological patterns of *Shigella* across SEA countries. The isolation rate of *S. flexneri* in Thailand started to decline in the mid-1990s, which echoes the trend reported by a recent human shigellosis surveillance study (from 1993 to 2006) performed in Thailand[12]. Similarly, the rapid increase of *S. sonnei* sequences

from Vietnam since the mid-2000s (Supplementary Fig. 2) corresponds with epidemiological observations from Southern Vietnam[13]. These data reflect independent replacement events of *Shigella* species occurring in these two well-sampled countries. Since shigellosis is not a notifiable disease in these countries, the samples are sourced from multiple diarrhoeal surveillance studies conducted in Southeast Asia in different time frames. Thus, the temporal trend in Supplementary Fig. 2 may not fully represent the epidemiological history of the pathogen.

Phylogenetic reconstruction of these data and globally sourced sequences demonstrated that the SEA *S. flexneri* samples isolated in the past three decades could be positioned in previously defined lineages (or phylogenetic groups, PGs), except for Lineages (Lin-) 5 and 6 (Fig. 2a). Based on the detection of specific combinations of the phage-borne serotype determining factors, *gtr* and *oac* variants, we overlaid the molecular serotype of all *S. flexneri* strains included in this study. In line with previous surveillance efforts, *S. flexneri* 2a (*n* = 431) was determined to be the predominant *S. flexneri* serotype in the region, followed by serotype 3a (*n* = 157)[12,13]. As apparent in Fig. 2a, Lin-3 (*N* = 606) is the dominant circulating lineage and is mainly composed of serotype 2a (*n* = 429/606). This is followed by Lin-2 (*N* = 143, serotype 3a exclusively) and Lin-1 (*N* = 110, serotypes 1a, 1b, 1c). Sub-lineages were divided further for Lin-3 and Lin-2. Lin-3 was split into three sub-lineages (3Xv, 3.1, 3.2), with 3.1 and 3.2 appearing to be of greatest epidemiological significance. Clear geographical clustering was observed for Lin-2, with Lin-2.1 and 2.2 comprised only of South and Southeast Asian organisms and 2p mainly endemic in Africa. In addition to known lineages, our data showed the presence of a novel lineage, consisting exclusively of isolates from SEA, and most closely related to Lin-7. We defined this as Lin-8.

The global phylogeny of *S. sonnei* reiterates that isolates from SEA mainly belong to the highly successful Lineage 3 (*N* = 765) (Fig. 2b). The whole genome phylogeny allowed delineation of this lineage into three sub-lineages. Two distinct sub-lineages were nested within Global 3 (the internationally disseminated major clone of *S. sonnei* Lin-3), and were named Global 3.1 (*N* = 455) and 3.2 (*N* = 202)[8]. These encompass the previously described *S. sonnei* expansions into Vietnam (IIIa and IIIb clones, respectively), with Global 3.1 shown to have undergone extensive clonal expansion across Vietnam[9]. The remaining sub-lineage, termed earlySEA (*N* = 101), circulated only in SEA, and was found to be most closely related to the South American expansion of *S. sonnei* in the 1950s[8]. A group of >30 SEA isolates were clustered within Lineage 1, which is unusual given that this lineage has only been associated with historical circulation of *S. sonnei* in Europe[8].

For consistency, we refer to the aforementioned lineages and sub-lineages of SEA *S. flexneri* and *S. sonnei* as "lineages" herein (i.e. *S. sonnei* Lineage Global 3.1), and monophyletic clades within these lineages as "clades".

**The evolutionary history and geographical spread of *Shigella* in Southeast Asia**. To enhance phylogenetic resolution and decipher the evolutionary history of SEA *S. flexneri* and *S. sonnei* within each lineage, we constructed individual maximum likelihood phylogenies for each examined lineage. All lineages, except for *S. flexneri* Lin-4, displayed a high correlation between sampling date and the phylogeny's root-to-tip divergence ($R^2$ ranging from 0.5 to 0.95; linear regression), thus indicating that most *Shigella* populations shows a clock-like single nucleotide polymorphism (SNP) accumulation rate (Supplementary Figs. 3, 4). Estimated nucleotide substitution rates were comparable among *Shigella* lineages ($5 \times 10^{-7}$ to $9 \times 10^{-7}$ substitutions per site per year) (Table 1), which is concordant with previous estimates from

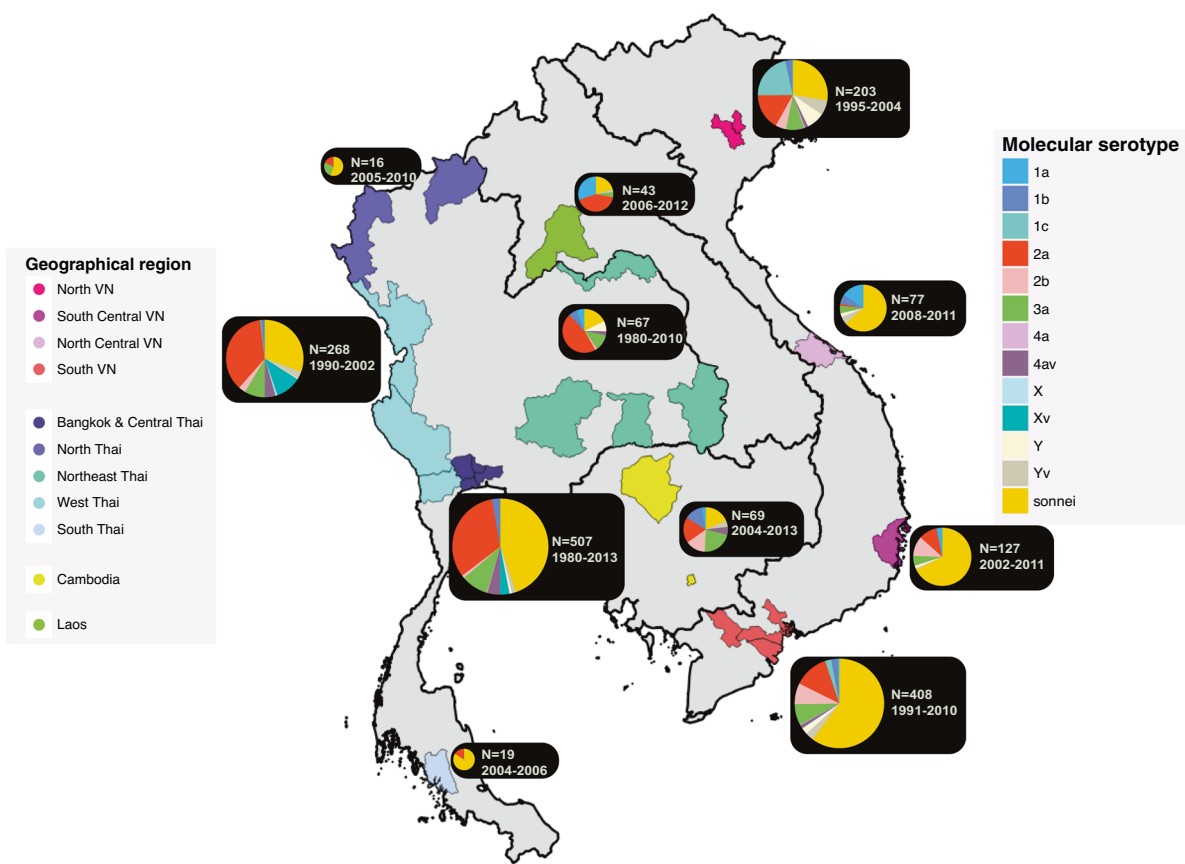

**Fig. 1 Overview of Southeast Asian *Shigella* genome sequences used in this study.** The figure displays a map of the four Southeast Asian countries where the *Shigella* samples originated (Thailand, Cambodia, Laos, Vietnam). Cities/provinces with collected Shigella sequences are highlighted with respect to the country's geographical divisions, except for Laos (where the capital, Vientiane, is highlighted to represent Lao isolates). These regions include Bangkok and Central Thailand (Bangkok, Nonthaburi, Samut Sakhon, Nakhon Pathom), North Thailand (Chiang Rai, Mae Hong Son), Northeast Thailand (Nong Khai, Ubon Ratchathani, Surin, Nakhon Ratchasima), West Thailand (Tak, Kanchanaburi, Ratchaburi), South Thailand (Trang), Cambodia (Phnom Penh, Siem Reap), North Vietnam (Hanoi), North Central Vietnam (Hue), South Central Vietnam (Khanh Hoa), South Vietnam (Ho Chi Minh City, Dong Thap, Tien Giang, and Ben Tre). The pie charts show the proportion of all *Shigella* molecular serotypes, as assessed by genomic analysis (see Methods), identified from the corresponding region (see key). The size of pie charts is proportional to the total number of isolates sequenced in the region. This is accompanied by a year range indicating the duration of sampling. The four countries' administrative maps (accessed via http://www.diva-gis.org/gdata) were merged to create a map of Southeast Asia, using QGIS v2.6.

global studies of *S. flexneri* and *S. sonnei*[8,10,14]. However, estimates for *S. flexneri* Lin-7 and Lin-8 and *S. sonnei* Lin-1 suggest that these organisms have significantly lower substitution rates ($1.4 \times 10^{-7}$ to $3.84 \times 10^{-7}$ substitutions per site per year). These lineages also formed the root of the *S. flexneri* and *S. sonnei* global phylogeny, respectively, and were genetically distant to their more rapidly disseminating counterparts. It has been shown in *S. sonnei* that an elevated evolutionary rate is associated with successful global spread and clonal establishment[8], which may facilitate adaptation in new environments. The slower molecular clocks of the aforementioned lineages (*S. flexneri* Lin-7 and Lin-8, *S. sonnei* Lin-1) may be associated with a corresponding reduction in fitness or adaptability, which may explain their lower than expected isolation rates in regional surveillance studies.

Using these data, we estimated the times to most recent common ancestor (TMRCAs) of multiple SEA *Shigella* lineages, inferred individually via Bayesian phylogenetics (BEAST v1.8.3) (Table 1). Despite their overarching genetic differences, the majority of these lineages appeared to have emerged within or entered the region during the last century, between the 1970s and 1980s (Supplementary Figs. 5–9). Some lineages, such as *S. sonnei* early SEA, *S. flexneri* 3Xv (serotype Xv) and *S. flexneri* 2.1 (serotype 3a), were composed of a single clone and largely

circulated exclusively within Thailand over a relatively short time period (<15 years from 1980 to 1995) with limited detection elsewhere in SE Asia (Supplementary Figs. 5–7). This suggests that these lineages could exhibit transient and restricted establishment in the region. In contrast, the phylogenies of other *Shigella* lineages showed a high degree of genetic diversity and a wider geographical and temporal distribution. *S. flexneri* Lin-3.1 and *S. sonnei* Global 3.1 contained exclusively SEA isolates, appearing as two substantial and extensive clonal expansion events that occurred throughout the region starting in the late 1970s (Supplementary Figs. 8, 9). Other lineages (*S. flexneri* Lin-1, 2.2, 3.2 and *S. sonnei* Global 3.2) in our dataset show wider geographical distributions, with SEA clades often interspersed with isolates from other continents. This observation indicates that these clades were likely introduced and established endemically circulating populations in SEA on three to five occasions (per lineage) after 1970. Once introduced, these lineages disseminated across the region (Supplementary Figs. 6–9). Geographical mapping conducted on lineages with highest recorded disease burdens (*S. flexneri* Lin-3.1 and 3.2, *S. sonnei* Global 3.1 and 3.2) demonstrated that the two species shared similar patterns of circulation in SEA (Supplementary Fig. 10). Intranational dissemination was frequent and

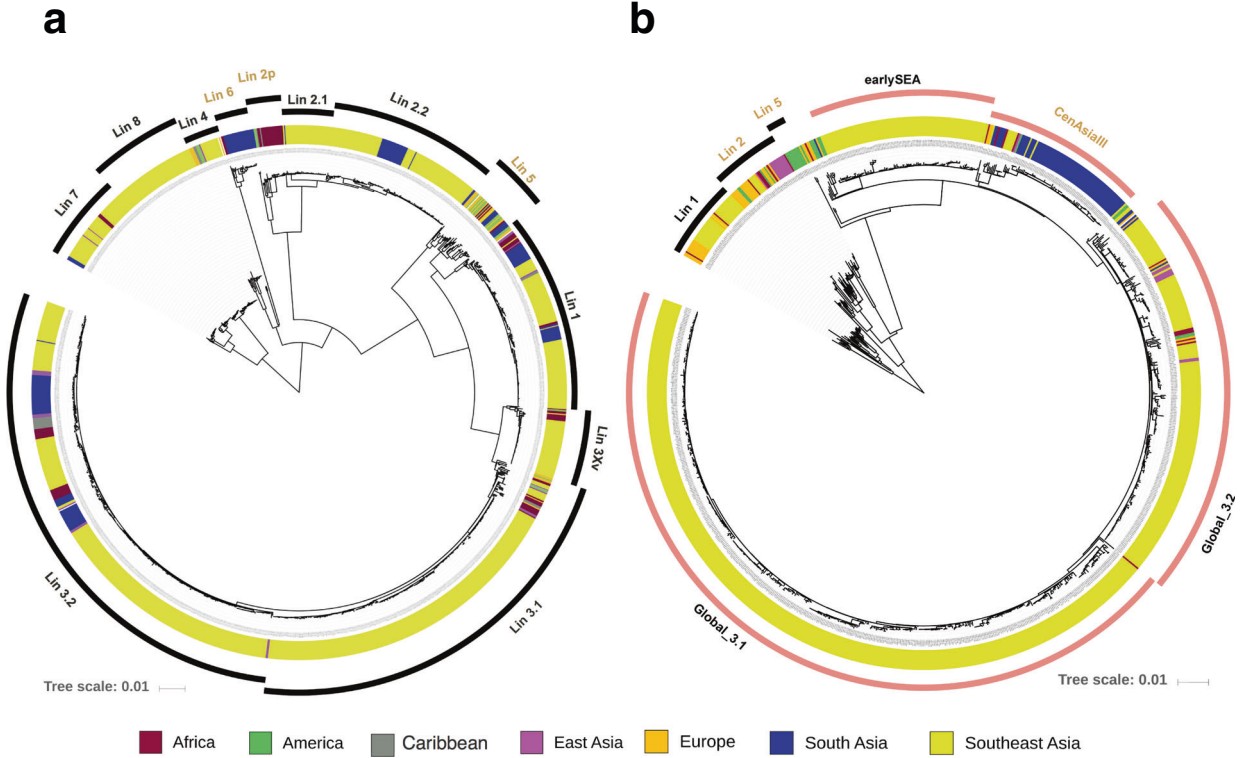

**Fig. 2 The global phylogenetic structure of *Shigella flexneri* and *Shigella sonnei*. a** The maximum likelihood phylogeny of 1323 *S. flexneri* isolates, compiled from those sequenced for this study and a previously published study on *S. flexneri* global diversity (ref. [10]). Different lineages were identified based on Bayesian Analysis of Population Structure hierarchical clustering (hierBAPS) and were named in accordance to the nomenclatures defined in ref. [10] (Lineages 1–7), with Lineage 8 newly defined in this study. The tree is rooted using Lineages 7 and 8 (see Methods). **b** The maximum likelihood phylogeny of 993 *S. sonnei* isolates, compiled from those sequenced for this study and previously published studies on *S. sonnei* global diversity (ref. [8]), *S. sonnei* local evolution in Vietnam (ref. [9]), and *S. sonnei* local evolution in Bhutan (ref. [34]). The isolates are grouped into lineages defined previously in the global diversity study (ref. [8]), and the tree is rooted using Lineage 1. For both (**a**) and (**b**), the colours of the inner ring indicate the isolate's geographical source (see key) while the outer ring defines the lineage clustering (with *S. sonnei* Lineage 3's sub-lineages coloured in pink). Lineages that do not contain substantial numbers of Southeast Asian sequences are labelled in orange and excluded from downstream analyses. The horizontal scale bar shows the number of nucleotide substitutions per site.

**Table 1 Summary of findings from Bayesian phylogenetic inference for major *Shigella* lineages.**

| Species | Lineage | TMRCA of SEA isolates (95% HPD lower – 95% HPD upper) | Substitution model | Clock model | Demographic model | Substitution rate (95% HPD lower – 95% HPD upper) |
|---|---|---|---|---|---|---|
| *Shigella flexneri* | 1 | 1974.43 (1972.36 – 1976.5) | TVM | Random local | BSk | 6.65 (6.1 – 7.25) x 10$^{-7}$ |
| | 2.1 | 1978.79 (1977.26 – 1980.57) | K3P | Random local | BSk | 7.58 (6.4 – 8.7) x 10$^{-7}$ |
| | 2.2 | 1948.69 (1942.15 – 1955.3) | TVM | Strict | BSk | 7.52 (6.7 – 8.31) x 10$^{-7}$ |
| | 3Xv | 1988.81 (1987.79 – 1990.04) | TVM | Random local | Constant | 8.60 (7.5 – 9.6) x 10$^{-7}$ |
| | 3.1 | 1978.66 (1977.74 – 1979.62) | TVM | Random local | BSk | 7.98 (7.4 – 8.56) x 10$^{-7}$ |
| | 3.2 | 1979.58 (1977.11 – 1982.29) | TVM | Random local | BSk | 8.85 (8.02 – 9.61) x 10$^{-7}$ |
| | 4 | NA | NA | NA | NA | NA |
| | 7 | 1400.09 (1236.37 – 1589.93) | TVM | Strict | Constant | 1.40 (1.00 – 1.78) x 10$^{-7}$ |
| | 8 | 1932.57 (1915.11 – 1953.65) | TVM | Random local | BSk | 3.66 (2.78 – 4.49) x 10$^{-7}$ |
| *Shigella sonnei* | Lin1 | 1869.93 (1834.41 – 1911.2) | TVM | Random local | Constant | 3.84 (3.04 – 4.63) x 10$^{-7}$ |
| | earlySEA | 1979.36 (1978.62 – 1980.19) | TVM | Random local | BSk | 8.89 (7.63 – 10.2) x 10$^{-7}$ |
| | Global 3.1 | 1975.95 (1972.07 – 1980.45) | TVM | Random local | BSk | 5.42 (4.81 – 6.04) x 10$^{-7}$ |
| | Global 3.2 | 1985.71 (1984 – 1986.05) | TVM | Random local | BSk | 7.99 (7.25 – 8.7) x 10$^{-7}$ |

The table details the estimation of major parameters, including the estimated TMRCA (time to most recent common ancestor) of isolates in Southeast Asia and its 95% high posterior density (95% HPD), the substitution rate (in substitutions per site per year) and its 95% HPD, as well as the molecular clock and demographic models utilised. For each lineage, the results are summarised across three independent BEAST runs, and the most appropriate evolutionary models were chosen based on Bayes factor comparison.
*TVM* Transversion model, *K3P* Kimura-3-parameters model, *BSk* Bayesian Skyline

characterised by *Shigella* transmission between major urban areas (i.e. Bangkok and Central Thailand; Southern Vietnam) and secondary population centres (Western and Northeastern Thai provinces; North-Central, and South-Central Vietnam), usually creating pan-regional *Shigella* populations. Specifically, inter-regional connectivity was most pronounced in *Shigella* lineages undergoing extensive clonal expansion across SEA (*S. flexneri* Lin-3.1, *S. sonnei* Global 3.1).

Further genomic interrogation found that serotype 4av was the most common, and likely the ancestral serotype for *S. flexneri* Lin-7 isolates (Supplementary Fig. 11). Frequent deletions of *gtrIV* and/or pSfYv prompt conversion to serotypes Yv, 4a, and Y. The TMRCA of Lin-7 was estimated to date back to the 15th century. However, circulations within SEA were founded by two major clades that likely emerged during the mid-1800s, with a lack of samples from outside of the region suggestive of in situ evolution and divergence of these two clades within SEA. *S. flexneri* Lin-8, for which the TMRCA was estimated to date back to the 1930s, is mostly comprised of serotypes Y and Yv (Supplementary Fig. 11). Previous surveillance studies have reported a low prevalence of these serotypes (4, Y, Yv) in the region, indicating the minor contributions of Lin-7 and Lin-8 to the clinical burden of shigellosis here[7,12,13]. Alternatively, *S. sonnei* Lin-1 was estimated to have been introduced to SEA in the mid-1800s but has subsequently been largely restricted to Northern Vietnam, with only occasional isolates seen elsewhere in SEA (Supplementary Fig. 5). The most closely related organisms, as well as the immediate sister group, to these SEA Lin-1 were historical organisms isolated in France from 1940s to 1970s.

**The antimicrobial resistance profile of *Shigella* in Southeast Asia.** The diversity and dynamic interplay of different *Shigella* lineages within SEA pose questions concerning the underlying factors that govern their ecology. We hypothesised that changes in AMR were associated with dynamics of *Shigella* populations. Since antimicrobial susceptibility testing was only available for a subset of isolates, we profiled the resistome of all isolates, supported by the consistent agreement between phenotypic and genotypic results for *Shigella*[15].

A high proportion of both *S. sonnei* and *S. flexneri* exhibited resistance to sulphonamides, tetracycline, and streptomycin (Fig. 3), evidencing the maintenance of these resistance types since as early as 1980. Resistance to these antimicrobials was conferred by a one-time acquisition of the spA plasmid (*sul2*, *tetRA*, *strAB*) in *S. sonnei* prior to the existence of the MRCA of CenAsiaIII, Global 3.1 and 3.2, but excluding earlySEA (Supplementary Fig. 12). Alternatively, earlySEA exclusively and stably harboured a genomic island that was almost identical to the *Shigella* Resistance Locus (SRL) (*aadA*, *bla*OXA-1, *tetB*, and *catA1*) previously reported in *S. dysenteriae* serotype 1[16]. These events explain the distinctive resistance profile of earlySEA among *S. sonnei* (Fig. 3b and Supplementary Fig. 12). The SRL island (*strAB*, *bla*OXA-1, *tetB*, and *catA1*) was also independently integrated into multiple *S. flexneri* lineages, except Lin-3Xv (Supplementary Fig. 13), in addition, granting *S. flexneri* ubiquitous resistance to ampicillin and chloramphenicol. The SRL was also subject to frequent modification, exemplified by the independent deletion of *bla*OXA-1 and *tetB* in Lin-2.2 around 1990 (Figs. 3a, 4c).

**Trimethoprim resistance coincides with clonal replacement in *Shigella*.** We observed a marked difference in resistance to trimethoprim pre- and post-1990 in *S. flexneri*, with the resistant genotype gradually becoming dominant in all populations (Fig. 3a). This apparent preference for trimethoprim resistance was evident in both the numbers of *dfrA* variants (eight genes) and the independent occasions upon which they were integrated into our *S. flexneri* collection (Supplementary Fig. 13). Specifically, the expansions of Lin-3.2, 3Xv, and part of 2.2 were all associated with the fixation of *dfrA1*, while independent acquisitions of a plasmid-borne *dfrA14* were pervasive in other lineages (Lin-1, 3.1, 7, 8). This distinctive *bla*TEM-1-*strAB-sul2-dfrA14*

AMR gene cluster was found to be co-transferred with a plasmid backbone highly similar to pRC960-1 (accession number: KY848295; 219/298 plasmid occasions), which was recently isolated from a *S. flexneri* Y of porcine-origin in China[17]. Independent acquisitions across multiple lineages in all examined countries resulted in the presence of the plasmid in ~300 *S. flexneri* isolates, commonly in organisms isolated from mid-1990s onwards. These results suggest a strong selective pressure favouring the maintenance of trimethoprim resistance in *S. flexneri*. We next sought to evaluate how *S. flexneri* population dynamics were altered in response to trimethoprim pressure. In order to limit the effects of pressure exerted by serotype-mediated immune escape, the analysis was conducted only for *S. flexneri* serotypes 2 (2a and 2b) and 3a. For serotypes 2 in Thailand and 3a across SEA, trimethoprim resistance increased rapidly in the 1980s and 1990s, nearly reaching fixation in both populations (Figs. 4a, 4c). This trend occurred concurrently with a shift in population structure, with Lin-3.1 replaced by Lin-3.2 (serotype 2), and Lin-2.1 replaced by Lin-2.2 (serotype 3a). Alternatively, an opposing trend was observed for serotype 2 in Vietnam, with trimethoprim resistance in gradual decline as Lin-3.1 began to predominate by 1996 (Fig. 4b).

Similarly, trimethoprim resistance accelerated dramatically in *S. sonnei* during 1980–1990, which occurred concurrently with the replacement of earlySEA by Global 3.2 and resulted in the predominance of the latter in SEA in the early 1990s. Trimethoprim resistance was mainly attributed to the stable inheritance of a class II integron (*dfrA1*) following its acquisition in the inferred MRCA of Global 3.1, 3.2, and CenAsiaIII (Supplementary Fig. 12). Moreover, other resistance mechanisms were also noted, including the integration of a class I integron (carrying *aph3-ereA-dfrA5*) into eight earlySEA isolates in 1989-90, and a 20 kbp MDR cassette (carrying *dfrA12-aadA2-qacEΔ-sul1-mphA-merADE- bla*TEM-1) into Lineage 1 in five separate events. These lines of evidence point to a preference for trimethoprim resistance in several *S. sonnei* lineages in the 1980s and 1990s. In contrast, the diminishing role of earlySEA resulted in the reduction of resistance to chloramphenicol and ampicillin across the region.

**Heightened resistance to quinolones and 3rd generation cephalosporin associated with recent endemic circulation in Southeast Asia.** Quinolone resistance in *Shigella* emerged in the mid-1990s and became more prevalent in the 2000s (Figs. 3, 4, 5) due to the proliferation of clones bearing mutations in *gyrA* (S83L or D87Y), which was apparent for *S. flexneri* lineages with widespread circulation in SEA (Lin-2.2, Lin-3.1, and to a lesser extent in Lin-8) (Figs. 3a, 4b, 4c). This finding mirrors previously published data for *S. sonnei* in Vietnam, in which the establishment of local populations of *S. sonnei* Global 3.1 was preceded by fixations of *gyrA* mutations[9].

The most pronounced difference between the resistomes of *S. flexneri* and *S. sonnei* was their susceptibility to 3rd generation cephalosporins. Resistance to this antimicrobial was predicted for only 6/1001 *S. flexneri* in our collection, with all *bla*CMY and *bla*CTX-M variants appearing from the year 2000 onwards (Supplementary Fig. 13). It is also clear that macrolide resistance emerged sporadically in *S. flexneri* over the entire period of the study with no apparent fixation (Fig. 3a). Conversely, carriage of several ESBL plasmids bearing *bla*CTX-M was noted in *S. sonnei* Global 3.1 isolated in Vietnam[9] (Fig. 5, Supplementary Fig. 12). However, in the early 2010s, the newly introduced fluoroquinolone resistant CenAsiaIII rapidly superseded Global 3.1 to be the most commonly detected[18], but this clonal replacement did not reverse the ongoing AMR trends. Alternatively, resistance to 3rd

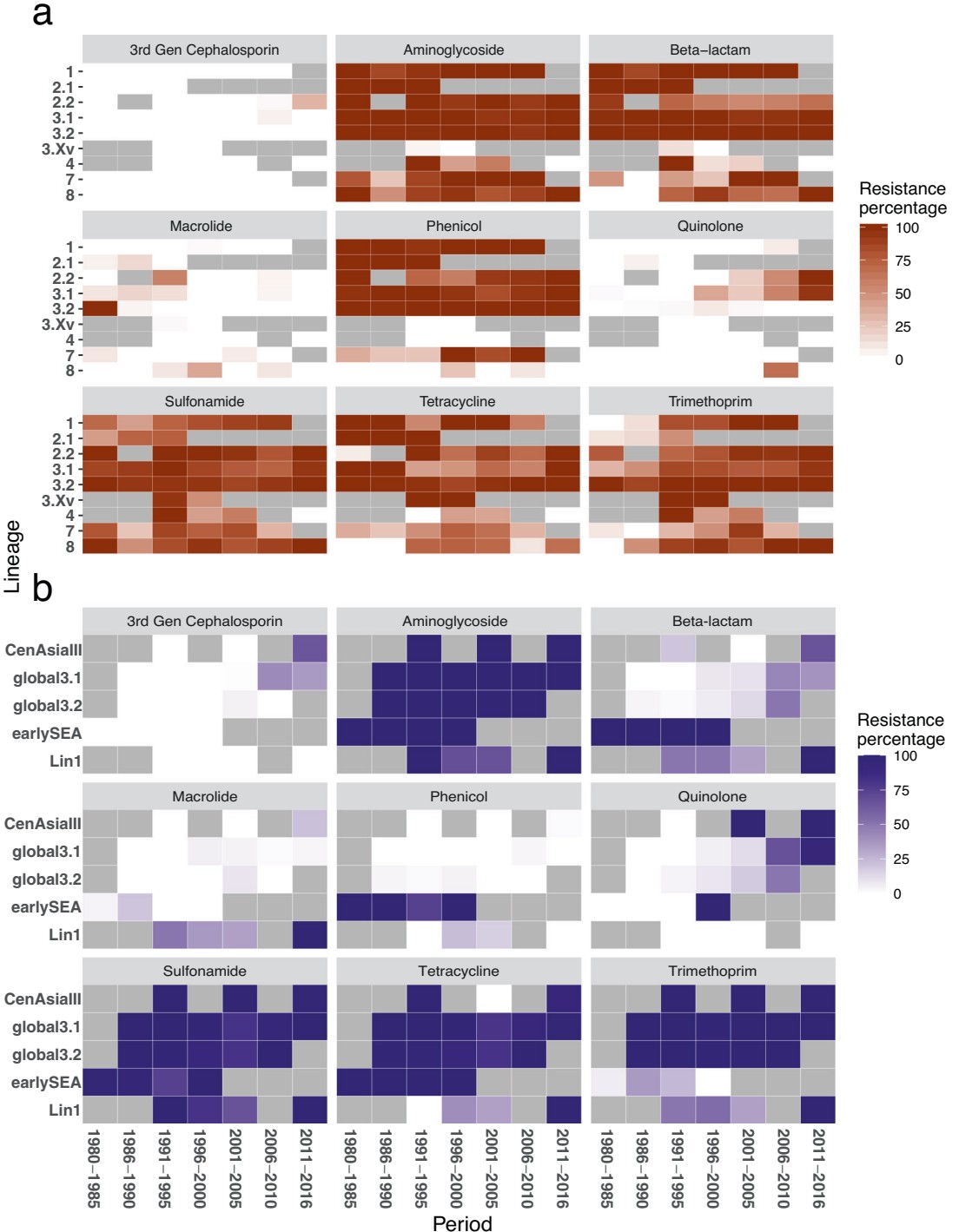

**Fig. 3 Trend of antimicrobial resistance (AMR) genotype in Southeast Asian _Shigella flexneri_ and _Shigella sonnei_. a** Trend of AMR genotype in 990 Southeast Asian _Shigella flexneri_ and (**b**) 880 Southeast Asian _Shigella sonnei_. Each heatmap plot details the resistance trend against a class of antimicrobial, as identified by presence of AMR genetic determinants (see Methods) and categorised into different phylogenetic lineages. The cell's colour intensity positively correlates to the proportion of sequenced isolates, belonging to a particular lineage during a defined duration, encoding resistance to the examined antimicrobial class (see key). The grey cells denote the absence of sequenced isolates.

generation cephalosporins and macrolides continued to increase, as evidenced by CenAsiaIII _S. sonnei_'s ability to sample the local AMR gene pool[19].

### Discussion

Our work represents the most comprehensive genomic epidemiology study of _Shigella_ in SEA to date. We employed an intensive sampling strategy to cover both temporal and spatial scales, resulting in a collection of >1800 genome sequences from four countries. Nevertheless, our interpretations remain subject to the uneven temporal distribution inherent to the collection. In particular, Thai isolates dominated from 1980 to 1995, while Vietnamese isolates were more common from 1995 onward. Organisms from Laos and Cambodia were scarce and only

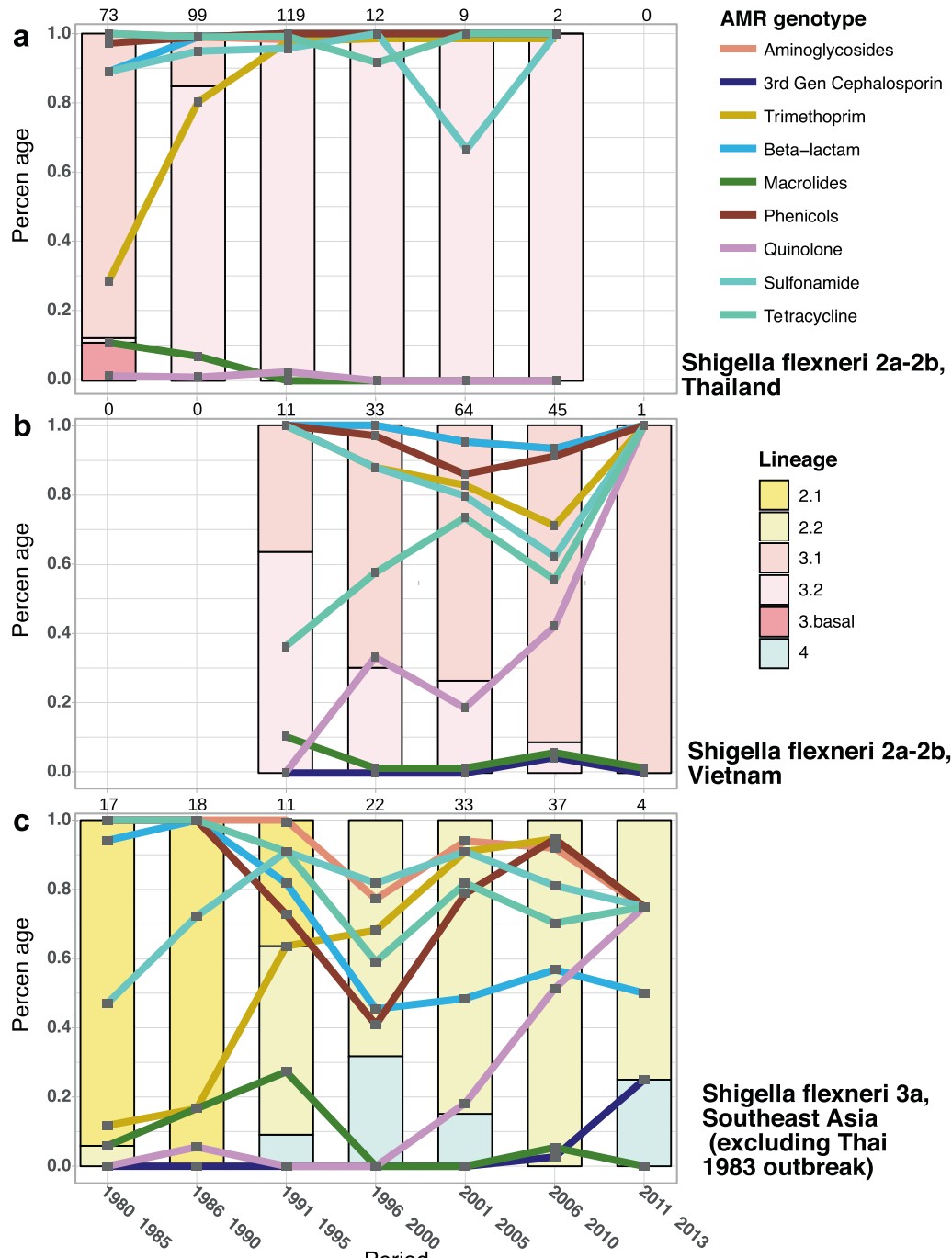

**Fig. 4 Clonal replacement of *Shigella flexneri* in Southeast Asia.** The figures display the isolation frequency of major *S. flexneri* lineages from 1980 to 2013, specifically for (**a**) serotypes 2 (2a, 2b) from all regions of Thailand, (**b**) serotypes 2 (2a, 2b) from all regions of Vietnam, and (**c**) serotype 3a from all countries across Southeast Asia (Thailand, Vietnam, Cambodia, Laos), excluding 11 Lin-2.2 isolates of a presumptive outbreak in Thailand in 1983 (see Method). The bar charts display the proportional sampling of these populations compared with the total *S. flexneri* of the respective serotype isolated during a period (total number of isolates indicated on top of each bar). The percentage of *S. flexneri* isolates resistant to a given antimicrobial, determined by identification of AMR genetic factors, is plotted by line graph. The y-axis (percentage) is used to measure both the isolation proportion of *S. flexneri* populations and proportion with resistance to specified antimicrobials.

available from the last decade. These limitations, together with the over-representation of major urban *Shigella* populations in this collection, may bias interpretations. Specifically, the predominance of Thai isolates in the earlier period might have favoured the inference of directional transmissions originating from this country. Further, our collection was limited to clinical samples from the paediatric population, ignoring potential *Shigella* diversity that may arise in environmental reservoirs

(for *S. flexneri*) and other shigellosis high-risk groups, such as the elderly or men-who-have-sex-with-men (MSM).

Notwithstanding these limitations, we were able to document the genetic diversity of various *Shigella* species and serotypes over the past three decades in SEA. *S. sonnei* was composed of several genetically distinct populations circulating over different time-frames. Within *S. flexneri*, a single serotype frequently consisted of several phylogenetically related but distinct clones, highlighting

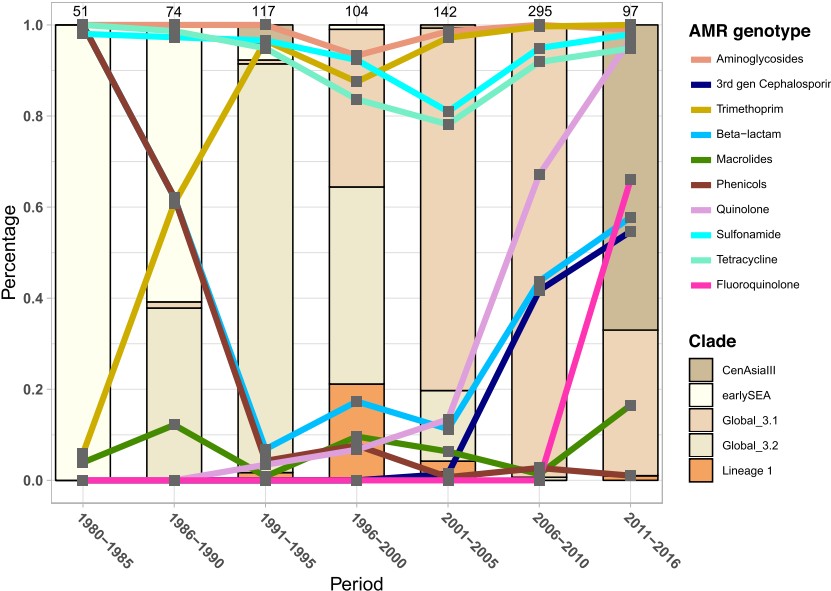

**Fig. 5 Clonal replacement of *Shigella sonnei* in Southeast Asia.** The figure shows the isolation frequency of *S. sonnei* lineages from 1980 to 2016 from all countries across Southeast Asia (Thailand, Vietnam, Cambodia, Laos). Isolates of Lineage 1 and earlySEA are mostly restricted to North Vietnam and Thailand, respectively. The bar chart displays the proportional sampling of these lineages compared with the total *S. sonnei* isolated during a period (total number of isolates indicated on top of each bar). The percentage of *S. sonnei* isolates resistant to a given antimicrobial, determined by identification of AMR genetic factors, is plotted by line graph. The *y*-axis (percentage) is used to measure both the isolation proportion of *S. sonnei* populations and proportion with resistance to specified antimicrobials.

the limitation of inferring the genetic makeup of the species from serotyping[10]. Despite the diverse genetic composition of multiple *Shigella* lineages, we found that the majority of their progenitors were likely introduced into continental SEA around 1970s-90s. Evidence for regional transmission of *Shigella* was most pronounced in lineages undergoing extensive clonal evolution, such as *S. flexneri* Lin-3.1 and *S. sonnei* Global 3.1, likely driven by increased human movement in the region. Indeed, domestic migration increased as economic growth accelerated and migration costs lowered in the 1990s, resulting in the relocation of 4.5 million people within Vietnam (~6% total population)[20]. Generally, we observed multiple introductions of *Shigella* lineages in the 20th century, except for two lineages (*S. sonnei* Lin-1 and *S. flexneri* Lin-7) having established regional endemic populations from the late 19th century. It is therefore intriguing that *S. sonnei* Lin-1 seems to have been maintained for more than a century, which stands in contrast to the relatively transient establishment of other *S. sonnei* populations observed in Southern Vietnam and Thailand. As this lineage is rarely found outside Europe, its distinctive epidemiology raises questions related to its exclusive success in Northern Vietnam. While most regions under consideration in these analyses (Central Thailand, Cambodia, Central and Southern Vietnam) experience a tropical climate throughout the year, Northern Vietnam belongs to the humid subtropical climate zone characterised by distinct seasonal temperature fluctuations. Meteorological factors have been associated with seasonal variation in shigellosis incidence in endemic countries[21]. Therefore, future research should examine how climatic differences affect the survival and transmission of *S. sonnei*, as well as the epidemiological significance of *S. sonnei* Lin-1.

Surveillance data have shown that *S. sonnei* is replacing *S. flexneri* as the major aetiological agent of shigellosis in most developing countries, including those in SEA[12,13,22]. However, there is little understanding of the intraspecific population dynamics of *S. sonnei* and *S. flexneri*. Importantly, our findings highlight the regularity of clonal replacement within this genus, as observed in both *S. sonnei* and major *S. flexneri* serotypes 2 and

3a. These ecological successions are often coupled with changes in resistance to contemporaneous antimicrobials, most notably to trimethoprim as outlined by resistome profiling in this study. This observation strongly suggests that response to antimicrobial usage is a key adaptive strategy in both species. Indeed, the deployment of co-trimoxazole for shigellosis treatment in the 1980s was quickly met with rapid increases in trimethoprim resistance by the mid-1980s across SEA[23]. Similarly, the current reliance on ciprofloxacin for shigellosis treatment has clearly instigated the global spread of fluoroquinolone resistant *S. sonnei*, which originated from South Asia around 2007[14]. In a separate scenario, clonal replacement has been noted in the recent shigellosis epidemiology of the MSM community in the United Kingdom. *S. flexneri* 3a was the predominant aetiology from the mid 2000s to early 2010s, and its propagation was linked with the acquisition of the pKSR100 plasmid encoding for resistance to azithromycin, an antimicrobial frequently prescribed for sexually transmitted infections in the MSM community[24]. Subsequently, the horizontal transfer of pKSR100 into other *Shigella* genotypes has similarly and repeatedly fuelled the emergence of novel variants in this population[25], culminating in the recent dominance of dual ciprofloxacin and azithromycin resistant *S. sonnei* (of CenAsiaIII sub-lineage) since 2017[26]. It is generally seen that intensified antimicrobial usage coupled with increased human mobility at the global scale may allow *Shigella* to exploit its high degree of genetic variability, at times favouring the establishment of newly introduced but highly resistant clones. Thus, changes in antimicrobial treatment in one locale could have far-reaching regional or global implications.

The epidemiology of *S. sonnei* is driven by intercontinental spread and recent local establishment of pandemic clones[8]; this trend is apparent in SEA, exemplified by the local expansion of Global 3.1 and multiple introductions of Global 3.2. In contrast, previous genomic epidemiology research has suggested that longer-term local establishment of lineages in endemic regions dominate the epidemiology of *S. flexneri* populations, in which a high degree of genetic diversity is frequently maintained in a

single location[10]. This notion is supported by the occasional recovery of environmental *S. flexneri* and the epidemiological association between this bacterial species and contaminated water[27,28]. Moreover, the virulence plasmid of *S. sonnei* is unstable at environmental temperatures, whereas *S. flexneri*'s counterpart is not[29]. *S. sonnei* also exclusively possesses a Type VI secretion system (T6SS) to overcome *E. coli*-established colonization resistance[30], as well as a plasmid-borne O-antigen and capsule granting resistance to complement-mediated killing[31], phagocytosis[32], and phagolysosomal degradation[33]. These lines of evidence suggest that *S. sonnei* may have adapted specifically for person-to-person transmission, while *S. flexneri* retains a more generalist lifestyle. However, our analysis offers a different perspective on *S. flexneri* evolution in SEA. While the population diversity of *S. flexneri* has been maintained by the simultaneous existence of multiple lineages, these were generally established from the 1950s onwards. This observation indicates that long-term circulation is uncommon for the majority of *S. flexneri* lineages in SEA. The emergence of new *S. flexneri* clones in the region could be attributed to either: (1) foreign introductions with or without local establishment or (2) successful expansions of specific clones from native *S. flexneri* populations. Recent clonal emergence and the admixture of regional and global isolates in a single lineage lend more support to the former postulation. Thus, we speculate that the introduction-local expansion mechanism supported the evolution of Lineages 1, 2.1, 2.2, 3Xv, 3.1, and 3.2, while the ancestral Lin-7 and Lin-8 represent a scenario where long-term local circulation may play a more prominent role. These results resonate with those recently described for *Vibrio cholerae*. In this pathogen, a single bacterial lineage responsible for the seventh cholera pandemic was shown to have been introduced into Africa on twelve distinct occasions, with the later and increasingly drug resistant waves replacing the pre-existing antimicrobial susceptible populations[34]. These introductions began in the 1970s, with the authors arguing that intense human migration and contact during the few decades of globalization resulted in inter- and intra-continental cholera transmission. This finding also downplays the role of aquatic *V. cholerae* in the pandemic's epidemiology. We speculate that although species diversity in both *S. flexneri* and *V. cholerae* are sustained via occupation of respective environmental niches, the majority of existent clinical burden could be attributed to expansions of specific clones, probably owing in part to increased human mobility and contact following globalization.

Collectively, our findings indicate that the evolutionary histories of both *S. flexneri* and *S. sonnei* in SEA were shaped by frequent clonal replacement events, which are linked to changing patterns of resistance to contemporaneous antimicrobials. However, antimicrobial resistance alone does not necessarily guarantee the continuing dominance of the *Shigella* species in the human population. The aetiological landscape of childhood diarrhoea in one of Vietnam's urban centres, for example, now mirrors that of developed settings, with decreasing isolation rates of *Shigella* and increasing non-typhoid *Salmonella* burden[35]. This is likely due to improved hygiene and sanitation, changing diet, or other factors tied to economic development. While our work focused on the impact of antimicrobial resistance, other factors could potentially contribute to the observed population dynamics in *Shigella*. These include stochasticity and population bottlenecks, as well as differences in population size or virulence potential at the sub-lineage level, which are difficult to quantify using only genomic data. Nevertheless, our work highlights the importance of continuous surveillance of shigellosis, preferably using high-resolution genomic techniques, to better inform public health measures and future intervention studies.

## Methods

**Organism collection and whole genome sequencing.** In order to explore the evolutionary history of *Shigella* in Southeast Asia, we gathered a collection of 1433 *Shigella* isolates from continental Southeast Asian countries, including Thailand, Vietnam, Laos and Cambodia. *Shigella* isolates were retrieved from multiple collaborative institutions in the region, including: Armed Forces Research Institute of Medical Sciences (AFRIMS) Thailand, $N = 1,239$; Lao-Oxford-Mahosot Hospital-Wellcome Trust Research Unit (LOMWRU), $N = 43$; Cambodia-Oxford Medical Research Unit (COMRU), $N = 19$; Institut Pasteur Cambodia, $N = 19$; Oxford University Clinical Research Unit (OUCRU) Ho Chi Minh City, $N = 113$. The bacterial isolates and data for this investigation originated from clinical studies approved by the scientific and ethical committees of The Hospital for Tropical Diseases (HTD) in Ho Chi Minh City, Vietnam, all other participating hospitals, the Institutional Review Board of the Walter Reed Army Institute of Research (Thailand data) and the Oxford Tropical Research Ethics Committee (OXTREC) in the United Kingdom. The study also included the characterization of bacterial isolates submitted for routine diagnostic purposes. Study participants or parents of young participants were required to provide written informed consent for the collection of samples and subsequent analyses, except when samples were collected as part of routine care. All of these *Shigella* were isolated from patients, predominantly young children, with dysenteric diarrhoea during diarrhoeal surveillance studies performed in the respective countries. For all of these isolates, genomic DNA was extracted from pure bacterial culture by staff scientists of the corresponding institutions, using the Wizard Genomic DNA Extraction Kit (Promega, Wisconsin, USA). DNA was stored at −20 ºC until shipment to the Wellcome Trust Sanger Institute (WTSI) for whole genome sequencing on an Illumina HiSeq2000 platform to produce paired-end reads of 125 bp in length. In addition, these data were appended with sequences available from previous studies, including 119 *S. flexneri* and 252 *S. sonnei* collected at OUCRU from previous diarrhoea surveillance research[9]. In summary, the compiled collection includes 1804 sequences (803 *S. sonnei* and 1001 *S. flexneri*), which covers multiple locations in the area, as detailed in Fig. 1 and Supplementary Table 1. These consist of *S. flexneri* of all reported phenotypic serotypes, except for serotype 6, as it is genetically more closely related to *S. boydii*. The collection times range from 1980 to 2013, but the timing of sample collections varies geographically.

**Shigella flexneri molecular serotyping.** Though *S. flexneri* serotyping was performed by collaborating institutes using commercial antisera (Denka Seiken, Japan), this was not available for all isolates and could not cover all *S. flexneri* epitopes arising from bacteriophage-mediated O-antigen modification. We sought to exhaustively identify serotype-determining genetic elements from *S. flexneri* whole genome sequences, adopting the methodology developed previously[36,37]. Genomic derived serotyping has shown a high concordance (>90%) with phenotypic serotyping, with the majority of discrepancies in serotype resulting from disruptive genetic mutations[15]. ARIBA v2.12.0 was used to identify serotype-specific determinants, including *gtrI*, *gtrII*, *gtrIV*, *gtrIc*, *gtrV*, *gtrX*, *oacA*, *oacB*, *oacD*, and *opt* (pSFXv_2). ARIBA outputs from each isolate were then compiled and curated. Genes that contained any deletion, frameshift, or nonsense mutations were deemed non-functional and counted as being absent from the isolate. Genomic serotype was defined based on specific combinations of these elements, as detailed in Supplementary Table 2.

**Short read mapping and phylogenetic reconstruction.** *S. sonnei* and *S. flexneri* sequencing reads were mapped to either the reference Ss046 (accession NC_007384) or Sf2a strain 301 (accession NC_004337), respectively, using the hash-indexing mapping tool SMALT (version 0.7.4)[38]. Candidate single nucleotide polymorphisms (SNPs) were called against the reference and filtered using SAM-tools v1.8 (mpileup option) and bcftools v1.8[39]. Low quality SNPs were removed if they met any of the following criteria: consensus quality <50, mapping quality <30, ratio of SNPs to read at a position <75%, read depth <4, or number of reads per strand <2.

Phylogenetic reconstructions were performed independently for *S. flexneri* and *S. sonnei*. To provide phylogenetic context, we combined the SEA *S. flexneri* collection with >300 sequences from the species global evolution study[10]. Regions pertaining to mobile and recombinogenic genetic elements were detected and removed by Gubbins v1.4.5[40], as were alignments columns with at least 2% indeterminate bases or gaps. This created a 51,616 bp SNP alignment across 1323 *S. flexneri* taxa, which was input into RAxML v8.2.4 for maximum likelihood phylogenetic reconstruction under the GTRCAT model, with 100 bootstrap replicates[41]. Phylogenetic groups (lineages) were defined as previously proposed, including Lineages 1, 2, 3, 4, 5, 6, and 7[10]. To identify the most appropriate root for the *S. flexneri* species tree, we constructed a maximum likelihood phylogeny based on a core genome SNP alignment of 111 representative sequences. These include ten random samples from each of the defined lineages (1, 2.1, 2.2, 3Xv, 3.1, 3.2, 4, 5, 6, 7, 8) and a *S. flexneri* serotype 6 isolate (phylogenetically closely related to *S. boydii*, and used as an outgroup). The core genome alignment was built using Roary v3.12.0, which utilised the annotations of de novo assemblies (see below)[42]. The resultant phylogeny indicated that the inferred common ancestor of Lineages 7

and 8 sits basal within the *S. flexneri* species phylogeny, so this inferred common ancestor was chosen as the most appropriate root. Maximum likelihood phylogenies were then generated independently in a similar manner for lineages containing substantial numbers of Southeast Asian sequences, including Lineages 1, 2, 3Xv, 3.1, 3.2, 4, 7, and 8.

For *S. sonnei*, we combined our SEA collection with 190 representative sequences from the species global evolution study[8,43]. Using the aforementioned procedures, a 14,241 bp SNP alignment of 993 sequences was generated and used for phylogenetic reconstruction in RAxML under a GTRCAT model, with 100 bootstrap replicates. Phylogenetic nomenclatures followed those established previously in the global study, and we subsequently constructed individual maximum likelihood phylogenies for the following groups: Lineage 1, Lineage 3 earlySEA, Lineage 3 Global 3.1, and Lineage 3 Global 3.2. A previous global study of *S. sonnei* showed that Lineage 1 sits basal within the species phylogeny[8]; thus, we rooted our *S. sonnei* phylogeny using Lineage 1. The *S. flexneri* and *S. sonnei* species global trees were visualised with associated metadata using the web-based interactive Tree of Life (iTOL) v4[44].

**Temporal structure analysis and Bayesian phylogenetic inferences**. We conducted temporal structure analyses and Bayesian phylogenetic inferences separately for each of the major *Shigella* lineages (*S. flexneri* Lineages 1, 2.1, 2.2, 3Xv, 3.1, 3.2, 4, 7, 8; *S. sonnei* Lineage 1, earlySEA, Global 3.1, and Global 3.2). To limit sampling bias, a maximum of five isolates per year per region were included for each lineage. Table 1 provides detailed information on the input for analysis of each lineage. Maximum likelihood phylogenies of the subset taxa were constructed independently for each lineage, following the aforementioned procedures. TempEst v1.5.1 was used to estimate the linear relationship between root-to-tip divergence of the input phylogeny and the isolates' attached date (sampling date in month/year)[45]. The most suitable substitution model for each input alignment was determined using the fast ModelFinder implemented in IQ-TREE v1.6.7[46,47]. BEAST v1.8.3 was used to infer evolutionary dynamics, including estimation of substitution rates and TMRCAs of SEA sequences for each *Shigella* lineage[48]. To identify the best-fit model for each dataset, multiple BEAST runs were conducted on combinations of the appropriate substitution model (Table 1), and a strict, random local or relaxed lognormal clock model in conjunction with a constant or Bayesian skyline demographic model. Each of these analyses was performed in triplicate using a continuous 50–200 million generation MCMC chain, with samples taken every 5000–20,000 generations, respectively. Tracer v1.6 was used to visually assess the parameter convergence of each run (ESS > 200). For robust model selection, both path sampling and stepping-stone sampling approaches were applied to each BEAST run to estimate marginal likelihoods[49,50], and the best model was selected based on comparison of Bayes factors. For the best model, triplicate runs were combined using LogCombiner v1.8.3, with removal of 20% burn-in.

**Stochastic mapping of geographical traits**. For major SEA *Shigella* lineages (*S. flexneri* Lin-3.1 and Lin-3.2; *S. sonnei* Global 3.1 and 3.2), we aimed to reconstruct the ancestral geographical states of each isolate using the maximum likelihood phylogeny, as previously described[14]. We treated geographical source (as documented in Supplementary Table 1) of the organisms as discrete characters. To control for biased sampling, for each lineage, we sub-sampled the input phylogeny to include an equal number of isolates ($N$ ranging from 11 to 18) from each character state, generating 500 subsampled trees. Stochastic mapping, implemented as the function make.simmap in the R package phytools v0.6.0, was applied to quantify transition events between geographical characters, separately for each subsampled tree[51]. The analysis was performed under an asymmetric model of character change (ARD) with the rate matrix sampled from the posterior probability distribution using MCMC (Q = mcmc) for 100 simulations[52].

**De novo assembly and determination of accessory genomes**. De novo sequence assembly for each isolate was performed using Velvet v1.2.03, with optimal k-mer size determined by VelvetOptimizer by varying its size between 66% and 90% of read length[53]. Sequence assemblies also followed the procedure described by the WTSI's Pathogen Genomic Bacteria Assembly pipeline, which facilitates robust and improved high-throughput prokaryote assembly[54]. Contigs with size <300 bp were removed, and read pairs were mapped back to the improved assembly using SMALT to assess the assembly quality. Annotation was determined for each assembly using Prokka[55]. ARIBA, an approach to detect genetic elements of choice from short reads, was implemented to identify the presence of antimicrobial (AMR) genes and plasmid incompatibility types, based on the reliably curated ResFinder and PlasmidFinder database, respectively[56–58]. The resulting resistome served as the isolates' AMR profiles for downstream analyses, supported by the consistent agreement between phenotypic and genotypic AMR findings for *Shigella*[15].

The resistance profile for each isolate was summarised based on the following detected determinants: 3rd generation cephalosporins ($bla_{CTX-M}$, $bla_{CMY}$, $bla_{DHA}$), aminoglycoside ($strAB$, $aadA$, $aac(3)$-IIa, $aph(3')$), beta-lactam ($bla_{TEM}$, $bla_{OXA-1}$), macrolide ($ermB$, $mefB$, $ereA$, $mphA$), phenicol ($cmlA1$, $floR$, $catA$, $cat3$), quinolone ($qnrS1$, $qnrB$, $gyrA$ codons S83 or D87 mutations), sulphonamide ($sul$), tetracycline

($tet$), trimethoprim ($dfrA$ variants), and fluoroquinolone ($gyrA$ codons S83 and D87, and $parC$ S80 mutations). The resistance profile for each antimicrobial was summarised for each *S. flexneri* and *S. sonnei* lineage, during a defined five- to six-year period (from 1980 to 2016). We additionally included 71 Southeast Asian CenAsiaIII lineage isolates in this analysis, sampled mostly from Vietnam after 2013, to provide an up-to-date overview on trends in AMR and clonal replacement. For analysis of *S. flexneri* serotype 3a, twelve Lin-2.2 isolates (all isolated from a hospital in Bangkok in 1983) were found to be genetically closely related (pairwise SNP difference ranging from 0 to 5). This suggests a presumptive shigellosis outbreak and is not representative of *Shigella* regional circulation. Thus, we only included one isolate from this 1983 outbreak to the AMR and clonal replacement analysis.

**Statistics and reproducibility**. General data summary, analysis, and visualization were conducted using R[59] (version 3.6.1) and RStudio[60] (version 1.1.383), including the package phytools v0.6.0[52]. Maximum likelihood phylogenies were constructed with at least 100 rapid bootstrap replicates, and Bayesian phylogenetic inference was performed in triplicate for each tested model specification. Stochastic mapping was performed on 500 subsampled phylogenies for each of the examined *Shigella* lineages.

**Reporting summary**. Further information on research design is available in the Nature Research Reporting Summary linked to this article.

## Data availability

Raw sequence data are available in the NCBI Sequence Read Archive (project PRJEB5281: Phylogeography of *Shigella spp.* in Southeast Asia). Data are available from the corresponding author on request. Source data for figures are provided in Supplementary Data 1.

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

## Acknowledgements

H.C.T. is a Wellcome International Training Fellow (218726/Z/19/Z). S.B. is a Wellcome Senior Research Fellow (215515/Z/19/Z). D.T.P. is a leadership fellow funded through the Oak Foundation. Whole genome sequencing was funded by S.B. and N.R.T. The work of LOMWRU was funded by Wellcome (grant number 106698/Z/14/Z), and we are grateful to all the patients and staff of the Microbiology Laboratory and wards who helped with the collection of isolates, to Bounthaphany Bounxouei, former Director of Mahosot Hospital; Chanphomma Vongsamphan, former director of the Department of Health Care, Ministry of Health; and Bounkong Syhavong, Minister of Health, Lao People's Democratic Republic, for their very kind help and support.

## Author contributions

H.C.T. performed data analysis and interpretation of the results under the scientific guidance of N.R.T., S.B., and M.A.R. H.C.T. drafted and edited the paper, with S.B. and M.A.R. revising and structuring the paper. L.B., D.T.P., C.J.M., T.H.T., P.V.V., P.T., S.H., D.A.B.D., P.N.N., V.D. and R.P. contributed to sample collection, storage, DNA extraction and sequencing. N.R.T. provided access to sequencing and bioinformatic analysis facilities. G.T., D.T.P., S.B., D.A.B.D., P.T. and N.R.T. contributed to the editing of the paper. All authors with the exception of RP (deceased) read and approved the final draft.

## Competing interests

The authors declare no competing interests.
