## [Peer Review File · Communications Biology]

Reviewers' comments:

Reviewer #1 (Remarks to the Author):

Chung The et al describe an important study of the population structure, antimicrobial resistance trends, and temporal, spatial and evolutionary dynamics of *Shigella flexneri* and *Shigella sonnei* in Southeast Asia. The authors describe the lineages/clades of both species in local, regional and global context, and dated the time to the most recent common ancestor for the extant lineages/clades in Southeast Asia. Crucially, the authors describe temporal changes in drug resistance of the isolates based on the genotypic data in context of the changes in the population structure. Overall, the study succeeds in its aim of showing that adaptation to antimicrobial pressure may have been pivotal to the evolutionary trajectory of *Shigella* in Southeast Asia.

The dataset analysed is large, the isolates spanned nearly three decades and the selection of representative isolates each year for whole genome sequencing was carefully done. The genomic analysis is robust and the authors seem to clearly understand the assumptions and limitations of the analytical methods employed particularly the BEAST and ancestral reconstruction analysis.

However, there are a few points that requires further clarification to improve the manuscript. The authors correctly pointed out that previous studies have suggested that the dominance of certain *S. sonnei* and *S. flexneri* lineages globally has been attributed to increased multi-drug resistance (MDR) and acquisition of virulence factors. However, the isolation rate of *Shigella* and distribution of certain lineages/clades for example *S. flexneri* lineage 3.2 appears to have decreased from early 1990s despite the increasing MDR rates. This appears to somewhat contradicts the hypothesis that adaptation to antimicrobials may have shaped the trajectories of the *Shigella* species in Southeast Asia. If MDR has been the main driver of *Shigella* population dynamics, why would there be a significant decrease in isolation rates when MDR rates seemed to be increasing? Are there any clinical interventions that may have contributed to the decrease in isolation rates? Could the antimicrobial adaptation have varying impact across different *Shigella* lineages which may explain why certain drug resistant lineages decreased over time or perhaps other factors including presence of virulence factors or possibly natural fluctuations of the lineages may have played an important role in shaping the dynamics of both species in Southeast Asia in addition to the MDR? The authors should clarify these statements and briefly mention in the discussion that there may be other factors in addition to MDR that may have shaped the population dynamics of the *Shigella* species in the study setting. Considering the potential role of virulence factors in shaping the dynamics of *Shigella* species as mentioned in the manuscript, the authors should consider conducting further analysis of the genomes to describe the distribution of known virulence factors to determine whether a combination of MDR and virulence may offer a better explanation of the trajectories seen in Southeast Asia for both species.

Minor comments:

1) The cited reference in line #52 does not suggest that *S. sonnei* species emerged nearly 400 years ago but rather shows the time of divergence for the most recent common ancestor of the modern-day strains in Europe. The actual time of emergence for the species is likely to be underestimated and much older than 400 years ago although it's unlikely that we may not know the actual time for sure as some of the lineages for the species that may existed long time ago may have become extinct prior to the expansion of the modern-day lineages.

- 2) Is there a reason why serotype 6 was excluded in the population structure analysis of *S. flexneri*? (line #59). The authors should clarify what's unique about the isolates belonging to this serotype relative to the other serotypes as it will help the reader to understand how the phylogeny of *S. flexneri* isolates was rooted later in the manuscript as isolates belonging to this serotype were considered to be an outgroup.
- 3) The authors state that accumulation of virulence factors may have driven the dominance of *S. flexneri* serotype 2a in lineage 3 (line #62). Considering that the virulence factors may play an important role in shaping the population structure and trajectories of the entire species or specific lineages, the authors should consider describing the distribution of key virulence factors in both species to complement the findings on drug resistance.
- 4) The isolation rate of *S. flexneri* and *S. sonnei* has decreased over time in Southeast Asia particularly Thailand (Fig. S1 and line #82) despite increasing drug resistance trends (Fig. 3 and 4). Some lineages including lineage 3.2 in Thailand which appear to have acquired drug resistance have decreased in frequency over time. This seems to be somewhat inconsistent with the main conclusion of the paper that adaptation to antimicrobials was pivotal to recent trajectory of *Shigella* in Southeast Asia. The authors should reconcile these somewhat contradictory statements.
- 5) Supplementary Fig. S2 shows that both *Shigella* species were only isolated close to the year 2010 in countries such as Cambodia and Laos. Does this imply recent emergence of these species in these countries or maybe it reflects a sampling issue whereby samples in the preceding years were not available for the analysis? The authors should clarify this in the relevant sections of the results as well as in the figure legend. It's also well-known that the number of isolates sampled per year may change depending on several factors for example more isolates may be collected when there is increased funding. The authors should include a statement regarding surveillance of these pathogens over time so that the reader can interpret the distribution of isolates shown in Fig. S1 accordingly. The authors should also consider including Fig. S1 in the main text as it provides important information on the distribution of the isolates overtime which may help interpretation of the incidence and drug resistance rates shown in Fig 3-5.
- 6) The choice of the serotype 6 strains as an outgroup for the *S. flexneri* phylogeny is not sufficiently justified. It would also be good to include a supplemental figure showing the resultant phylogeny described in line #403 which may help the reader to see visually how the rest of the *S. flexneri* lineages cluster with the outgroup serotype 6 isolates. Additionally, the authors should similarly clarify how the phylogeny of the *S. sonnei* was rooted.
- 7) The authors should explicitly state whether the drug resistance profiles were determined solely using genotypic data (which I assume was the case) or whether antimicrobial susceptibility tests were also done. It's not clear in the methods whether the latter was done.

Reviewer #2 (Remarks to the Author):

Chung The et al. present a very interesting work focusing on the evolutionary dynamics of the two most important *Shigella* species in Southeast Asia. The manuscript is very well written, the methods are robust, the results and discussion sections are straightforward and easy to follow, limitations of this work are accurately acknowledged, and the figures are quite elucidative. I guess this is the first I have no major concerns to report, so I would like to take this opportunity to congratulate the authors for a fine piece of research! I would recommend moving figures S12 and S13 to the main manuscript, but I'm OK if the authors decide to keep them as supplementary. Also, most tools in the material and methods sections are missing the used version (for example, ARIBA, Gubbins, RAxML, Roary, Prokka).

Best wishes,
João Botelho

Referee expertise:

Referee #1: Pathogen genomics, pathogen evolution

Referee #2: Antibiotic resistance, mobile genetic elements, bioinformatics

Reviewers' comments:

Reviewer #1 (Remarks to the Author):

Chung The et al describe an important study of the population structure, antimicrobial resistance trends, and temporal, spatial and evolutionary dynamics of *Shigella flexneri* and *Shigella sonnei* in Southeast Asia. The authors describe the lineages/clades of both species in local, regional and global context, and dated the time to the most recent common ancestor for the extant lineages/clades in Southeast Asia. Crucially, the authors describe temporal changes in drug resistance of the isolates based on the genotypic data in context of the changes in the population structure. Overall, the study succeeds in its aim of showing that adaptation to antimicrobial pressure may have been pivotal to the evolutionary trajectory of *Shigella* in Southeast Asia.

The dataset analysed is large, the isolates spanned nearly three decades and the selection of representative isolates each year for whole genome sequencing was carefully done. The genomic analysis is robust and the authors seem to clearly understand the assumptions and limitations of the analytical methods employed particularly the BEAST and ancestral reconstruction analysis.

However, there are a few points that requires further clarification to improve the manuscript. The authors correctly pointed out that previous studies have suggested that the dominance of certain *S. sonnei* and *S. flexneri* lineages globally has been attributed to increased multi-drug resistance (MDR) and acquisition of virulence factors. However, the isolation rate of *Shigella* and distribution of certain lineages/clades for example *S. flexneri* lineage 3.2 appears to have decreased from early 1990s despite the increasing MDR rates. This appears to somewhat contradicts the hypothesis that adaptation to antimicrobials may have shaped the trajectories of the *Shigella* species in Southeast Asia. If MDR has been the main driver of *Shigella* population dynamics, why would there be a significant decrease in isolation rates when MDR rates seemed to be increasing? Are there any clinical interventions that may have contributed to the decrease in isolation rates?

We thank the reviewer for these insightful comments. Our analyses showed that resistance to contemporaneous antimicrobials may have played a role in driving intraspecific clonal replacement events in Shigella, but antimicrobial resistance alone does not necessarily guarantee the continuing dominance of Shigella in the population. Countries in SEA have undergone rapid economic transition, and development has coincided with decreasing rate of Shigella isolation in childhood diarrhoea. This is probably owing to improved hygiene and sanitation, increased access

to clean drinking water and healthcare, other public health measures or other unknown factors. We have added more on this in the last paragraph of the Discussion, specifically “Collectively, our findings indicate that the evolutionary histories of both S. flexneri and S. sonnei in SEA were shaped by frequent clonal replacement events, which are linked to changing patterns of resistance to contemporaneous antimicrobials. However, antimicrobial resistance alone does not necessarily guarantee the continuing dominance of the Shigella species in the human population. The aetiological landscape of childhood diarrhoea in one of Vietnam’s urban centres, for example, now mirrors that of developed settings, with decreasing isolation rates of Shigella and increasing non-typhoid Salmonella burden³⁵. This is likely due to improved hygiene and sanitation, changing diet, or other factors tied to economic development.” (page 15, lines 346-353)

Could the antimicrobial adaptation have varying impact across different Shigella lineages which may explain why certain drug resistant lineages decreased over time or perhaps other factors including presence of virulence factors or possibly natural fluctuations of the lineages may have played an important role in shaping the dynamics of both species in Southeast Asia in addition to the MDR? The authors should clarify these statements and briefly mention in the discussion that there may be other factors in addition to MDR that may have shaped the population dynamics of the Shigella species in the study setting. Considering the potential role of virulence factors in shaping the dynamics of Shigella species as mentioned in the manuscript, the authors should consider conducting further analysis of the genomes to describe the distribution of known virulence factors to determine whether a combination of MDR and virulence may offer a better explanation of the trajectories seen in Southeast Asia for both species.

We thank the reviewer for their suggestions. We agree that there are other factors which could contribute to the shaping of population dynamics in Shigella. These could be stochastic (natural fluctuation, population bottleneck), differing population sizes, or differing virulence potentials. We focused on antimicrobial resistance in this manuscript as it has shown similar effect in other pathogens (i.e. Vibrio cholerae) and is effectively quantified from genomic data. We opted not to include detailed analysis on virulence as (1) it could not be effectively quantified from genomic data, particularly for S. sonnei and a portion of S. flexneri Lineage 1 (the large virulence plasmid is unstable, particularly in storage, and is not retrievable from sequencing), (2) inference of phenotypic virulence from genomic data is unreliable for Shigella (except for the Shiga toxin encoding stx, which was not present in all of our SEA isolates), (3) its inclusion would greatly lengthen our manuscript and dilute the discussion on antimicrobial resistance. We have acknowledged the potential contribution of other factors, including virulence in the Discussion: “While our work focused on the impact of antimicrobial resistance, other factors could potentially contribute to the observed population dynamics in Shigella. These include stochasticity and population bottlenecks, as well as differences in population size or virulence potential at the sub-lineage level, which are difficult to quantify using only genomic data.” (pages 15-16, lines 353-358)

Minor comments:

1) The cited reference in line #52 does not suggest that *S. sonnei* species emerged nearly 400 years ago but rather shows the time of divergence for the most recent common ancestor of the modern-day strains in Europe. The actual time of emergence for the species is likely to be underestimated and much older than 400 years ago although it's unlikely that we may not know the actual time for sure as some of the lineages for the species that may existed long time ago may have become extinct prior to the expansion of the modern-day lineages.

We agree with the reviewer's comment on the time to most recent common ancestor of modern S. sonnei. We have amended the sentence as "An early key genomic study on the global collection of extant S. sonnei revealed that the contemporary variant of this pathogen likely emerged in Europe at least 400 years ago." (page 3, lines 52-54)

2) Is there a reason why serotype 6 was excluded in the population structure analysis of *S. flexneri*? (line #59). The authors should clarify what's unique about the isolates belonging to this serotype relative to the other serotypes as it will help the reader to understand how the phylogeny of *S. flexneri* isolates was rooted later in the manuscript as isolates belonging to this serotype were considered to be an outgroup.

S. flexneri serotype 6 in fact shares a close phylogenetic relationship with other serotypes of S. boydii, and is distantly related to the remaining S. flexneri serotypes on the phylogenetic tree. Therefore, S. flexneri 6 is not included in analyses (this and those previously reported) of S. flexneri species evolution. We have now clarified this in the manuscript by stating it explicitly "excluding serotype 6 (which is phylogenetically closely related to Shigella boydii serotypes) (reference 11)." (page 3, lines 60-61)

3) The authors state that accumulation of virulence factors may have driven the dominance of *S. flexneri* serotype 2a in lineage 3 (line #62). Considering that the virulence factors may play an important role in shaping the population structure and trajectories of the entire species or specific lineages, the authors should consider describing the distribution of key virulence factors in both species to complement the findings on drug resistance.

As addressed above.

4) The isolation rate of *S. flexneri* and *S. sonnei* has decreased over time in Southeast Asia particularly Thailand (Fig. S1 and line #82) despite increasing drug resistance trends (Fig. 3 and 4).

Some lineages including lineage 3.2 in Thailand which appear to have acquired drug resistance have decreased in frequency over time. This seems to be somewhat inconsistent with the main conclusion of the paper that adaptation to antimicrobials was pivotal to recent trajectory of *Shigella* in Southeast Asia. The authors should reconcile these somewhat contradictory statements.

As addressed above.

5) Supplementary Fig. S2 shows that both *Shigella* species were only isolated close to the year 2010 in countries such as Cambodia and Laos. Does this imply recent emergence of these species in these countries or maybe it reflects a sampling issue whereby samples in the preceding years were not available for the analysis? The authors should clarify this in the relevant sections of the results as well as in the figure legend. It's also well-known that the number of isolates sampled per year may change depending on several factors for example more isolates may be collected when there is increased funding. The authors should include a statement regarding surveillance of these pathogens over time so that the reader can interpret the distribution of isolates shown in Fig. S1 accordingly. The authors should also consider including Fig. S1 in the main text as it provides important information on the distribution of the isolates overtime which may help interpretation of the incidence and drug resistance rates shown in Fig 3-5.

Since shigellosis is not a notifiable disease in these four Southeast Asian countries (Cambodia, Laos, Thailand, Vietnam), the isolates included in this study are not expected to fully represent the epidemiological history of the pathogen. The sequences included in our study are sourced from multiple diarrhoeal surveillance studies, as described in the Methods, and samples from Laos and Cambodia in particular were only available in the period when these studies were conducted. We acknowledge that our interpretations remain subject to inherent sampling bias, and this has been noted in the first paragraph of the Discussion. We have added in the Results "Since shigellosis is not a notifiable disease in these countries, the samples are sourced from multiple diarrhoeal surveillance studies conducted in Southeast Asia in different time frames. Thus, the temporal trend in Figure S2 may not fully represent the epidemiological history of the pathogen." For Fig S2' figure legend, we have included the statement "Samples were sourced from multiple diarrhoea surveillance studies". Fig. S1 demonstrates that our sequencing effort matched the epidemiological patterns of Shigella in Bangkok and Central Thailand, but it does not necessarily capture the epidemiological trend of Shigella across the region as whole. In our opinion, Fig. S1 is not a main result, and we retain it in the supplementary information.

6) The choice of the serotype 6 strains as an outgroup for the *S. flexneri* phylogeny is not sufficiently justified. It would also be good to include a supplemental figure showing the resultant phylogeny described in line #403 which may help the reader to see visually how the rest of the *S. flexneri* lineages cluster with the outgroup serotype 6 isolates. Additionally, the authors should similarly clarify how the phylogeny of the *S. sonnei* was rooted.

As addressed above, S. flexneri serotype 6 is phylogenetically distantly related to the other S. flexneri serotypes, which warrants its use as an outgroup. We did not include the resultant phylogeny described in line #403 since the long branches separating serotype 6 and the other lineages made it hard to visualize. Instead, this phylogeny showed that the most recent common ancestor for Lineages 7 and 8 diverged prior to the diversification of other known true S. flexneri lineages. We amended Methods to read “The resultant phylogeny indicated that the inferred common ancestor of Lineages 7 and 8 sits basal within the S. flexneri species phylogeny, so this inferred common ancestor was chosen as the most appropriate root.” (page 19, lines 414-416) For S. sonnei, we included additional information for clarity “A previous global study of S. sonnei showed that Lineage 1 sits basal within the species phylogeny; thus we rooted our S. sonnei phylogeny using Lineage 1.” (page 19, lines 425-427)

7) The authors should explicitly state whether the drug resistance profiles were determined solely using genotypic data (which I assume was the case) or whether antimicrobial susceptibility tests were also done. It's not clear in the methods whether the latter was done.

We have included the sentence “The resulting resistome served as the isolates’ AMR profiles for downstream analyses, supported by the consistent agreement between phenotypic and genotypic AMR findings for Shigella” in the Methods section for clarity, since AMR phenotypic data were not available for all isolates. (page 21, lines 472-474)

Reviewer #2 (Remarks to the Author):

Chung The et al. present a very interesting work focusing on the evolutionary dynamics of the two most important Shigella species in Southeast Asia. The manuscript is very well written, the methods are robust, the results and discussion sections are straightforward and easy to follow, limitations of this work are accurately acknowledged, and the figures are quite elucidative. I guess this is the first I have no major concerns to report, so I would like to take this opportunity to congratulate the authors for a fine piece of research! I would recommend moving figures S12 and S13 to the main manuscript, but I'm OK if the authors decide to keep them as supplementary. Also, most tools in the material and methods sections are missing the used version (for example, ARIBA, Gubbins, RAxML, Roary, Prokka).

We thank the reviewer for their time, consideration and encouraging comments on our manuscript. Regarding figures S12 and S13, we think the AMR genotyping information has been succinctly summarized in Figure 3, and would like to retain these in the supplementary information. As suggested by the reviewer, we have added the version used for the bioinformatic tools described in the Methods section.

REVIEWERS' COMMENTS:

Reviewer #1 (Remarks to the Author):

The authors have fully addressed my previous comments. I have no further comments.

Reviewer #2 (Remarks to the Author):

The authors have properly addressed the comments from both reviewers.